# Classical and Modern Prejudice toward Individuals with Intellectual Disabilities: The Role of Experienced Contact, Beliefs in a Just World and Social Dominance Orientation

**DOI:** 10.3390/ijerph21030355

**Published:** 2024-03-16

**Authors:** Rocco Servidio, Ivan Giuseppe Cammarata, Costanza Scaffidi Abbate, Stefano Boca

**Affiliations:** 1Department of Culture, Education and Society, University of Calabria, 87036 Arcavacata di Rende, Italy; 2Department of Psychology, Educational Science and Human Movement, University of Palermo, 90128 Palermo, Italy; ivangiuseppe.cammarata@unipa.it (I.G.C.); costanza.scaffidi@unipa.it (C.S.A.); stefano.boca@unipa.it (S.B.)

**Keywords:** beliefs in a just world, classical prejudice, intellectual disabilities, modern prejudice, social dominance orientation

## Abstract

This study investigates the impact of experienced contact on prejudiced attitudes towards individuals with intellectual disabilities (IDs), examining beliefs in a just world (BJW) and social dominance orientation (SDO) as potential serial mediators. Data were collected from 224 university students (M = 23.02, SD = 2.48). Path analysis modelling assessed the structural relationships between the study variables. The findings revealed that experienced contact was negatively and significantly associated with BJW and SDO. Additionally, BJW and SDO fully mediated the relationship between experienced contact and overt prejudice. These findings underscore the influence of individual differences on attitudes towards individuals with ID, establishing a crucial foundation for future research and the development of interventions aimed at reducing prejudice and discrimination.

## 1. Introduction

According to Schalock and colleagues [1], intellectual disabilities (IDs) are characterised by severe limits in cognitive functions and adaptive behaviour across a wide variety of everyday tasks evident before age 18. Neurodevelopmental disorders like ID are classified as “Intellectual and Developmental Disabilities (IDD)” in the most recent international classifications, along with conditions like Autism Spectrum Disorders, Attention Hyperactivity Disorder, Learning Disabilities, Motor Developmental Disabilities, and Communication Disorders [2]. Despite the widespread deinstitutionalisation of individuals with IDs today, societal attitudes remain sluggish in shedding the negative labels of “hopeless cases” and “a problematic population” that have been attached to them for so long [3]. These social perceptions can significantly impact the level of societal inclusion and community involvement for individuals with IDs, affecting their overall quality of life and well-being [4,5,6]. Therefore, it is crucial to grasp the prevailing social attitudes towards IDs to pinpoint areas that require attention in awareness campaigns and educational initiatives [7].

Indeed, prior studies indicate that, despite advancements in understanding and accepting various forms of disabilities, people with IDs are frequently the target of stigma and prejudice (e.g., [4,8,9,10]). Furthermore, other studies have highlighted that individuals with ID constitute one of the most marginalised and stigmatised groups in society (e.g., [4,11]). Overcoming prejudice towards individuals with ID represents a significant challenge to achieving a truly inclusive and equitable society [12].

Negative attitudes towards disabled individuals could impede their social participation and integration [7,11,13,14,15]. Moreover, prejudice towards individuals with ID remains a significant societal issue, hindering their social inclusion and overall personal well-being [6]. This underscores the importance of fostering an inclusive society, aligning with the fundamental principles of the Sustainable Development Goals (SDGs). The SDGs provide a framework for global cooperation and action to address the world’s most pressing challenges in a holistic and integrated manner. The SDGs are a set of 17 global objectives adopted by the United Nations in 2015 as part of the 2030 Agenda for Sustainable Development. Each SDG is interconnected, and progress in one goal often depends on progress in others. They build upon the successes and shortcomings of the Millennium Development Goals (MDGs) and aim to address a broader range of interconnected issues, including poverty, inequality, climate change, environmental degradation, peace and justice. It involves promoting positive contact between individuals with and without ID and challenging negative stereotypes through media campaigns and advocacy strategies.

Prejudice describes a tendency to evaluate members of out-groups as less than in some way than in-group members [16]. It is a viewpoint shaped by personal beliefs and pre-existing ideas, lacking a genuine understanding of the relevant facts and individuals. In recent years, scholars have started differentiating between classical (e.g., old-fashioned, blatant, overt) and modern (e.g., subtle, covert) manifestations of prejudice. More specifically, classical forms involve direct or open prejudice, while modern forms are characterised by their covert or subtle nature (e.g., [16,17]). According to Sears [18], modern prejudice is characterised by three components: denial of continued discrimination, antagonism towards minority group demands and resentment about special favours for minority groups. Therefore, while classical prejudices are often directed to race, ethnicity and religion (e.g., racial segregation, antisemitism), modern prejudices (e.g., paternalism) encompass a broader range of identities and characteristics, reflecting the complexities of contemporary societies [19]. This twofold conceptualisation of prejudice has been adopted in various social and cultural contexts for different types of discrimination towards minority groups [20] as well as towards people affected by ID [21].

This study aims to contribute to a better understanding of the role of the variables associated with negative attitudes towards people with ID. Negative attitudes perpetuate prejudice, such as perceiving individuals with ID as helpless, burdensome or incapable of making meaningful contributions [22]. Understanding the underlying causes and factors contributing to prejudice against individuals with ID is crucial for developing effective interventions [7].

Research suggests that ignorance, fear and a lack of personal experience with individuals affected by ID play significant roles in perpetuating prejudice against members of minority social groups [21,23,24,25]. Additionally, societal factors, such as media portrayal and cultural biases, can influence social and personal attitudes towards individuals with ID. Prejudice towards individuals with ID, in turn, has far-reaching consequences for their well-being and social inclusion. Empirical studies demonstrate that experiences of prejudice contribute to lower self-esteem, increased social exclusion, limited educational and employment opportunities, and compromised mental health among individuals with IDs [5,10,26] compared to their typically developed peers. Efforts to combat prejudice towards individuals with ID have focused on several key areas, including education, awareness campaigns, inclusive policies and fostering positive intergroup contact [27,28]. However, the intergroup contact hypothesis proposed by Allport [16] represents a promising approach to reducing prejudice toward minority groups, such as persons with ID. According to this strategy, increasing personal and positive contact often improves views towards members of negatively stereotyped groups, but casual encounters are more likely to reinforce prejudice than to dispel it, in contrast to personal and meaningful connections [20,27,29]. Allport [16] suggests that to create the best conditions for attitude improvement, interpersonal interaction involving a cooperative and dependent connection supporting status equity and opposing stereotypes is required [30]. Moreover, it seems that the association between intergroup contact and positive attitudes is often influenced by disability type. For instance, Huskin and colleagues [29] discovered a connection between social distance and regular contact: more contact with those with mental illness resulted in fewer feelings of social distance. For sensory deficits, no such association has been discovered. Another study found that the nature of the contact was a significant predictor of students’ attitudes towards people affected by disabilities, suggesting the importance of the quality of the contact [27].

Therefore, our goal was to expand existing research on factors systematically associated with negative attitudes towards individuals with ID. Specifically, we examined the combined influence of experienced contact, belief in a just world (BJW) and social dominance orientation (SDO) to explore their effects on prejudice towards people with ID. Therefore, individuals who socialise in contexts where beliefs in a just world are prominent—the tendency to believe that adverse events befall those who bear responsibility—tend to avoid adopting the social responsibility norm [31]. It is akin to making a deal with the world, where people believe that if they are good and do the right things, good things will happen to them, and they will achieve their personal and social goals [for a review, see [32]]. These beliefs help people feel safe and secure, giving them hope for the future.

For example, concerning prejudice, individuals who score high in BJW show a positive relationship with negative attitudes towards individuals with mental illness [33]. Additionally, it has been found that BJW is associated with harsh social attitudes and dominance [34]. It appears that a convincing belief in a just world can lead individuals to develop explanatory theories characterised, for instance, as ‘conspiratorial attitudes’ and ‘self-directed’ when attempting to make sense of social phenomena. This tendency not only impacts people’s mental well-being, including their subjective comfort and levels of anxiety [35], but also influences their social relationships. For instance, it may affect their interpersonal sensitivity and ability to empathise and understand others’ emotions. This can be particularly evident when considering individuals who belong to minority groups, such as people with disabilities. BJW represents a fundamental component of an individual’s personality that significantly influences their behaviours and experiences. Indeed, people high in BJW may tend to have discriminatory attitudes and may experience discomfort in interactions with people affected by ID, aligning with the notion that individuals receive outcomes based on what they deserve [for a review, see [35]]. Therefore, individuals who have faith that the world is a just place may be more motivated to attribute blame to marginalised social groups to maintain their worldview beliefs that individuals get what they deserve [36].

Social dominance theory provides a framework for understanding societies’ hierarchical structure and maintaining group-based inequalities [37]. Social dominance orientation (SDO; [38]) has been studied in terms of prejudice towards individuals with ID, shedding light on how attitudes towards social hierarchies may influence biases and discriminatory behaviours directed towards this population [17,33,39,40]. Research has shown that individuals with high SDO endorse and support the idea of a fixed social hierarchy, exhibiting more prejudice towards lower-status minorities, such as individuals with ID [41]. They engage in discriminatory behaviours or hold negative stereotypes about ID-affected people’s abilities and worth, increasing social distance. Individuals who score high in SDO may view persons with ID as less deserving of equal opportunities, resources and social inclusion, reinforcing discriminatory attitudes and behaviours. Moreover, Bäckström and Björklund [42] found that individuals with higher scores in SDO were more predisposed to demonstrate classical and modern prejudice towards individuals perceived to have “impaired development” compared to those with lower scores in this orientation. Similarly, Brandes and Crowson [7] examined the relationship between conservative ideologies (e.g., SDO) and discomfort with disability among preservice educators. They found that SDO and discomfort with disability were stronger predictors of negative attitudes towards people with disabilities and opposition to social inclusion. In another study, Crowson and Brandes [13] found that individuals high on SDO and right-wing authoritarianism (RWA) were more likely to reject rights for persons with intellectual and physical disabilities than individuals scoring lower on these variables.

De Keersmaecker and Roets [43] found that belief in a just world (BJW) and SDO tend to be correlated; people who have a strong belief in a just world endorse and support a rigid hierarchical structure in society, accepting and even promoting inequality among social groups. Moreover, Oldmeadow and Fiske [44] have characterised BJW and SDO as forms of ideologies related to system justification, specifically relevant to social inequality. They argue that these variables play a role in shaping status stereotypes to justify social inequality, suggesting that biases against impoverished individuals are influenced by factors that justify the existing social system. Therefore, we anticipated that BJW and SDO would be directly associated with negative attitudes. Individuals who strongly believe that the world systematically compensates for the good and evil one does are also likely to score high in prejudice towards individuals with ID. Additionally, those who envision the social world as a hierarchy of groups rigidly determined by their worthiness are expected to manifest higher prejudice. Furthermore, we hypothesised that BJW and SDO would mediate the association between experienced contact and prejudice. BJW refers to individuals’ tendency to believe that people generally get what they deserve [45].

### The Current Study

This study aimed to investigate the attitudes of able-bodied students towards individuals affected by ID. We also aimed to identify underlying factors that could either heighten or attenuate negative attitudes towards them. Therefore, the primary objective of the present study was to test a model in which experienced contact serves as the predictor, with BJW and SDO as potential serial mediators and prejudice toward individuals with ID as the outcome. Notably, previous studies have not simultaneously investigated the role of experienced contact and the two individual difference variables (BJW and SDO) and their correlation with classical and modern prejudice against people with ID. This suggests a significant research gap regarding the impact of experienced contact on these relationships.

Therefore, the current study intended to examine the relationship between experienced contact and the combination of two individual difference variables (BJW and SDO) and the expression of prejudice (classical and modern) against people affected by ID. Given that individuals with ID may face beliefs of inadequacy, it is argued that SDO and BJW serve as systems legitimisation factors for social inequality. Consequently, a positive correlation is expected between these factors and prejudice towards individuals affected by ID. Specifically, based on the results of the discussed studies, we predicted that experienced contact should have a buffering role and should reduce negative attitudes towards people with ID.

Drawing upon Duckitt’s [46] theory of ideology and prejudice, we hypothesised that prejudice stems from enduring ideologies rooted in social beliefs, known as worldviews. SDO is identified as a robust predictor of prejudiced attitudes, and we predict that BJW acts as an antecedent to SDO. Individuals with fewer opportunities for positive intergroup contact are expected to exhibit high levels of both BJW and SDO, justifying their prejudicial attitudes towards people with ID [47]. 

Then, based on the limited findings of previous studies (e.g., [17,33]), we hypothesised that individuals with fewer opportunities for positive intergroup contact might exhibit high levels of BJW (H_1_) and high levels of SDO (H_2_) as a strategy to justify their prejudicial attitudes towards people with ID. Moreover, given the results of prior studies (e.g., [7,41]), we expected (H_3_) that SDO rather than BJW should have a strong impact on prejudice towards people affected by ID. Additionally, we hypothesised (H_4_) that the opportunity for intergroup contact should decrease both BJW and SDO, and these variables may serially mediate the relationship between the experience of contact and prejudice towards people with ID. For example, the interaction between individual differences and diversity (e.g., people affected by ID) is expected to diminish the effects of the mediator variables (BJW and SDO), thereby decreasing the levels of prejudice toward individuals affected by ID. Understanding the mediating role of SDO and the effect of BJW allows for designing targeted intervention programs that promote positive contact experiences and challenge negative beliefs and attitudes toward individuals with ID, which could be helpful.

## 2. Materials and Methods

### 2.1. Participants and Procedures

A sample of 224 Italian participants (62 males and 162 females) was recruited for this study using a snowball procedure. Initially, a class of university students was contacted, and they were asked to fill in the online questionnaire and to share the link with their friends and mates. The participants’ ages ranged from 19 to 33 years (M = 23.02, SD = 2.48). The participants were informed that the study aimed to investigate social attitudes towards people affected by intellectual disabilities. When asked to report their university degree course, participants identified as follows: 158 (70.5%) belonged to a social sciences degree course (i.e., pedagogy, psychology, etc.), and 66 (29.5%) were in the Science, Technology, Engineering, and Mathematics (STEM) field (i.e., mathematics, engineering, etc.). All of the participants were recruited from the University of Calabria and the University of Palermo (both located in southern Italy) during break periods and before the beginning of classes. The participants were given a thorough introduction outlining the study’s objectives before being asked to fill out an online questionnaire. According to the ethical standards, the researcher stressed that participation was entirely voluntary, allowing students to withdraw their participation at any point. Furthermore, the participants were promised complete anonymity and reassured that their data would be used solely for research purposes. No compensation or extra university credits were provided to participants. Upon obtaining consent from the students, those interested were given a questionnaire that typically required around 20 min to complete. The data were collected following the ethical standards of the Italian Psychological Association (AIP) and adhering to the Helsinki Declaration [48] and its later amendments.

### 2.2. Measures

When a validated Italian version of the scale was unavailable, a forward and backward translation approach was adopted to ensure the preservation of the original meaning of the items.

#### 2.2.1. Social Dominance Orientation

The Italian version of the social dominance orientation (SDO) 10-item scale [38,49] was used. This scale includes items such as “*Some groups of people are simply inferior to other groups*”. The participants were required to indicate how much they agreed with each item on a 4-point Likert-type scale of 1 (*strongly disagree*) to 4 (*strongly agree*). Cronbach’s alpha showed acceptable internal consistency of the scale α = 0.63.

#### 2.2.2. Beliefs in a Just World

A short version of the Lipkus [50] scale was translated and adapted from English into Italian and used to measure beliefs in a just world (BJW). The scale includes seven items; an example of an item is “*I am confident that justice always triumphs over injustice*”. Each participant had to rate their level of agreement on a 7-point Likert-type scale from 1 (*strongly disagree*) to 7 (*strongly agree*). The value of the Cronbach α was good, 0.81.

#### 2.2.3. Modern and Classical Prejudices Scale

Prejudice towards individuals with intellectual disabilities was measured with the Italian Modern and Classical Prejudices Scale (MCPS-IT) [21,23]. The MCPS is a 19-item scale for prejudice towards people with ID. It investigates two forms of prejudice: classical or overt/direct (e.g., *People with intellectual disabilities often commit crimes*) and modern or covert/subtle (e.g., *People with intellectual disabilities are getting too demanding in their push for equal rights*). The participants responded on a 5-point Likert-type scale ranging from 1 (*Strongly disagree*) to 5 (*Strongly agree*). The internal reliability was α = 0.73 for classical prejudice and α = 0.74 for modern prejudice.

#### 2.2.4. Experienced Contact as a Predictor Variable

The predictor variable measured in this study [adapted from Voci and Hewstone [51]] was a composite index computed as the product of (a) quality of contact with the out-group and (b) quantity of contact with the out-group. Quality of contact was measured with four items preceded by, “*When you meet a person affected by intellectual disabilities, in general, do you find the contact* …”. The four quality items, assessed on a 5-point Likert-type scale from 1 (*never*) to 5 (*often*), were “cooperative”, “superficial”, “voluntary”, and “equal” (Cronbach’s α = 0.66). The quantity of contact was assessed with a single item rated on a 5-point Likert-type scale from 1 (*never*) to 5 (*very frequently*): “*How frequently do you have contact with a person affected by intellectual disabilities*?” According to Voci and Hewston [51], in order to obtain a single index of frequent and positive intergroup contact, we multiplied the scores of quality and quantity of contact. This approach, as utilised by Brown and colleagues [52], enables us to simultaneously account for both dimensions of contact. Indeed, focusing solely on either the quantity or quality of contact is frequently insufficient in mitigating prejudice; a blend of the two is optimal, as highlighted by Allport [16]. By following Rozich and colleagues [53], before multiplication, quality scores were also re-coded so that –2 indicated negative contact and +2 positive contact. The quantity scores were re-coded so that 0 corresponded to no contact experience and 3 to highly frequent contact. Therefore, the composite index ranged from –6, indicating a high amount of negative contact, to +6, indicating a high amount of positive contact. The midpoint 0 may denote either the absence of contact or the presence of a neutrally valenced intergroup contact [51].

#### 2.2.5. Demographic Profile

All of the participants completed a socio-demographic profile collecting information about their gender, age and degree course.

## 3. Data Analysis

All of the statistical analyses were carried out with the support of Jamovi (version 2.3) software [54] and R-lavaan (version 0.6-13) package [55]. The missing data were not an issue, as the online questionnaire required participants to input responses for any missed items. Initially, we inspected the reliability of administered measures using Cronbach’s alpha (α) coefficient. Consequently, descriptive statistics, univariate normality and Pearson’s r correlations were run to explore the variables’ properties and determine their relationships. The scale scores were normally distributed [56]. To verify the hypothesised relationships between the variables, we designed and tested a path model with observed variables. Specifically, experienced contact was the predictor, BJW and SDO were the mediators and classical and modern prejudice were the outcomes. Direct paths were estimated from the predictor to the outcome variables, as well as correlations between the control, mediators and outcome variables. Furthermore, a bootstrapping procedure using 5000 resamples was performed to test the significance of indirect effects with a 95% confidence interval (CI) [57]. In estimating all path coefficients, we controlled for gender since it can influence people with ID [27]. 

Finally, to assess the adequacy of the proposed research model, we decided to explore a second-trimmed alternative model. The trimmed model included constraining non-significant paths to zero. This approach allows us to determine how well the proposed model aligns with the observed data.

## 4. Results

### 4.1. Descriptive and Correlations

The results of descriptive statistics among study variables are shown in Table 1.

We found statistically significant correlations in the hypothesised directions between the predictor variable (experienced contact), SDO, BJW and attitudes (classical and modern prejudice) (Table 2). Classical prejudice was positively related to SDO, *r*(224) = 0.53, *p* < 0.001, and BJW, *r*(224) = 0.37, *p* < 0.001. At the same time, modern prejudice was only associated with BJW, *r*(224) = 0.40, *p* < 0.001. A positive association between SDO and BJW, *r*(224) = 0.36, *p* < 0.001 was also found.

### 4.2. Mediation Analysis

The bootstrapped regression-based path analysis displayed direct and indirect effects. The results of the estimated direct effects for the saturated model are shown in Figure 1.

All of the associations in the model were statistically significant, except for the following paths: (a) from experienced contact to classical prejudice, (b) from experienced contact to modern prejudice, and (c) from SDO to modern prejudice. Experienced contact negatively and significantly predicted BJW and SDO. Both BJW and SDO positively and significantly predicted classical prejudice. BJW was a significant positive predictor of classical and modern prejudice, respectively. The total indirect effect (IE) of experienced contact on classical prejudice was statistically significant, Β = −0.04, 95% CI [−0.07, −0.02], β = −0.27, *p* < 0.001. Specifically, we found that BJW and SDO were serial mediators in the relationship between experienced contact and classical prejudice, Β = −0.04, 95% CI [−0.01, −0.01], β = −0.02, *p* < 0.01. The current results suggest that experienced contact was indirectly associated with classical prejudice through BJW, Β = −0.01, 95% CI [−0.00, −0.01], β = −0.04, *p* < 0.05, and SDO, Β = −0.02, 95% CI [−0.03, −0.01], β = −0.12, *p* < 0.01, respectively. No other significant results emerged from the current analyses. Regarding the control variables, gender negatively affected only classical prejudice, Β = −0.13, 95% CI [−0.23, −0.04], β = −0.14, *p* < 0.05. Classical prejudice accounts for 34% of the variance. 

Given the cross-sectional design of the study, we ran a second alternative model in which we trimmed the non-significant paths from the first model. Specifically, we tested an additional model by fixing non-significant paths to zero (e.g., dashed lines in Figure 1). The results indicated that the total indirect effect of experienced contact with classical prejudice once again was statistically significant, Β = −0.03, 95% CI [−0.05, −0.02], β = −0.20, *p* < 0.001. Experienced contact was a negative and significant predictor of BJW and SDO, respectively. In turn, BJW was associated with classical and modern prejudice as well as with SDO (*p* < 0.001). SDO was a significant predictor of classical prejudice (*p* < 0.001). The results of the alternative model (see Figure 2) suggest that experienced contact was indirectly associated with classical prejudice through the serial mediation of BJW and SDO as well as by considering the specific effects of BJW and SDO.

## 5. Discussion

This study sought to investigate the serial mediating effects of BJW and SDO in the relationship between experienced contact and classical and modern prejudice towards individuals affected by ID. Building upon prior research on this phenomenon and drawing from the literature on prejudice and individual differences, the results of the current study provide substantial support for the assumption that experienced contact is an important factor in reducing prejudice, mainly classical prejudice, against people affected by ID. Pearson’s correlations reveal that BJW is correlated with modern prejudice, while SDO is associated with classical prejudice against people with ID.

The key findings of this study revealed that experienced contact was negatively associated with both BJW (H_1_ supported) and SDO (H_2_ supported), and these mediating variables significantly influenced classical prejudice (but not modern prejudice) toward individuals with ID. These results suggest that higher-quality intergroup interactions are associated with lower levels of these cluster ideological variables, suggesting that positive intergroup contact can play a pivotal role in reducing prejudice and mitigating the effects of BJW and SDO in discriminating against people with ID. However, the direct association between the experienced contact and prejudice (classical and modern) was not statistically significant.

One of the most notable findings of this study was the relationship between SDO and prejudice against people with ID (H_3_ supported). In line with previous studies, SDO, which reflects a desire for hierarchical intergroup relations and social inequality, was found to have a stronger effect on classical prejudice than modern prejudice towards people affected by ID [7,14,15]. In other words, this result suggests that university students who express discomfort with disability are more likely to perceive society hierarchically and favour social inequality. Therefore, classical prejudice, characterised by overt and explicit forms of bias, may be more closely linked to SDO than modern prejudice, which tends to manifest subtly and less overtly. These findings are consistent with previous research linking SDO to prejudice against socially subordinate groups [51], such as people affected by ID [11,17]. This result suggests the complexity of prejudice and the need for nuanced approaches to understanding its underlying mechanisms.

Along with SDO, the current results suggest that individuals who scored higher in BJW, characterised by the belief in a fundamentally just and fair world where people generally receive what they deserve, were more likely to express prejudice (classical and modern). This finding highlights the significant role of BJW in explaining the relationship between certain individual factors (e.g., individual characteristics, social influences) and the expression of prejudice. In the context of people affected by ID, BJW can lead to the assumption that people with ID must have done something to deserve their condition or circumstances. These beliefs can be particularly harmful, as they may justify discrimination against people with ID. Unlike overt forms, modern prejudice is often more subtle and can manifest through microaggressions, implicit biases or veiled negative attitudes towards specific groups, such as individuals affected by ID.

Moreover, our bootstrapped regression-based path analyses suggest that the experienced contact with classical prejudice (but not modern prejudice) toward people with ID is fully mediated through the serial effects of BJW and SDO (H_4_ partially supported). Additionally, experienced contact was indirectly related to classical prejudice through the mediating role of BJW and SDO, respectively. Overall, it appears that reduced experienced contact with individuals affected by ID increases people’s tendency to view the world in terms of social hierarchies, and exhibiting preferences for one’s group to dominate over others is linked to prejudice against persons with ID. This connection is demonstrated by a diminished motivation to actively strive against prejudicial responses toward those groups and a distinct willingness to resist or limit their rights. The results of this study not only support but also extend previous research [20,27,29] by considering the role of two individual differences, such as BJW and SDO, which have never been examined in this field, and using modern prejudice as an outcome variable. Including these variables allows for a more comprehensive exploration of the intricate dynamics that underlie intergroup relations and prejudice against people affected by ID.

The current results are partially consistent with the hypotheses of the study since the experienced contact influences BJW, which impacts SDO and, in turn, affects classical prejudice but not modern prejudice. Further, our results indicated a full serial indirect effect of BJW and SDO in the relationship between experienced contact and classical prejudice. Taken together, the current study provides valuable insights into the underlying mechanisms driving prejudice towards individuals with ID and underscores the importance of understanding the complex interplay of beliefs, social hierarchies and attitudes in intergroup contexts. Individuals with higher levels of BJW and SDO tend to support legitimising myths that justify group-based inequality. Moreover, in line with the theory of ideology and prejudice [46], people who endorse these system-justifying ideologies might increase the likelihood of adopting social policy attitudes that reinforce the existing status differences within society [41] for whom they already hold prejudice for members of minority groups (e.g., students with ID) [6]. Notably, the relationship between these variables persisted in the alternative revised model, suggesting how individuals with ID are affected by negative attitudes.

Finally, in this study, the differentiation between modern and classical prejudice offers new insights into research exploring the risk of discrimination towards people affected by ID. Modern prejudice, characterised as a more ‘sophisticated’ form, demands increased cognitive effort and justification. The current results are consistent with the initial study by Akrami and colleagues [17]. Therefore, this study seeks to distinguish between classical and modern forms of prejudice, shedding light on the complexity of discriminatory attitudes.

### Limitations and Further Studies

The current study has several limitations. First, all the variables were measured using self-report questionnaires. This method is susceptible to social desirability bias, where participants may respond in a way they believe is socially acceptable, even if it is not their valid response. Future studies should use various methods to measure prejudice, such as implicit measures, to reduce the impact of social desirability bias. Second, the study only examined the serial mediating role of BJW and SDO in the relationship between experienced contact and prejudice. It is possible that other variables, such as empathy or perceived similarity, may also play a role in this relationship. Future studies should investigate these other variables to gain a more complete understanding of the mediation process. Third, the study was conducted with a sample of university students in southern Italy. This limits the ability to generalise the findings to other populations. Future studies should replicate the findings with a more diverse sample of participants. However, the current sample is relevant to the objective of the study since most of them will work with people who could be affected by ID, and other studies interviewed university students in similar research [21]. Furthermore, additional studies should gather information about the students’ profiles and university degree programmes, focusing on educational subjects related to individuals affected by ID. Finally, the study was cross-sectional, meaning that the data were collected at one point in time. This makes it difficult to conclude the causal relationships between the variables. Future studies should use a longitudinal design to track participants and better understand the causal mechanisms involved. Following this line of discussion, the limitation concerning the issue of causality should also be mentioned. In structural models, causal relationships between factors are often assumed a priori and are not falsified by the data, even if the true causal relationship is the reverse of that being suggested [42]. However, it should be noted that in the current study, we integrated into a single model social factors (experienced contact) and personality factors (SDO and BJW) as suggested by scholars [58,59].

## 6. Conclusions

Overall, the present study suggests that individuals with higher levels of experienced contact show less antisocial behaviour and should be more open to interacting with people affected by ID. This study highlights the relevance of individual differences in understanding prejudice against people with ID. The mediating role of BJW and SDO, in the relationship between experienced contact and prejudice, should help those responsible for social and inclusive policies in designing targeted intervention programs to promote the social inclusion of individuals with ID disorders.

## Figures and Tables

**Figure 1 ijerph-21-00355-f001:**
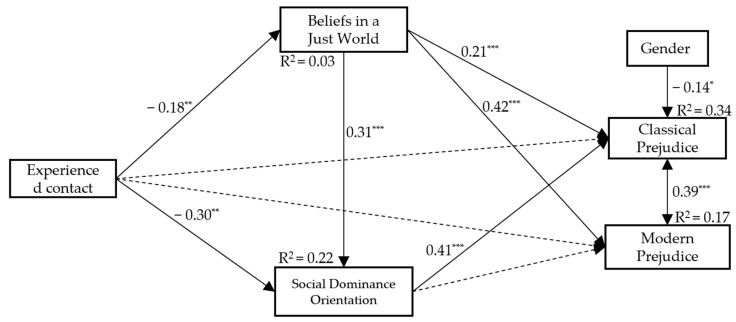
Standardised path coefficients for the whole model. Dashed lines indicate non-significant paths. Note. Gender (1 = male; 2 = female). * *p* < 0.05. ** *p* < 0.01. *** *p* < 0.001.

**Figure 2 ijerph-21-00355-f002:**
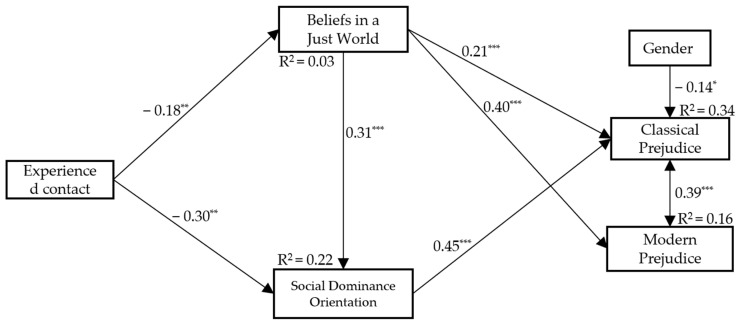
Standardised path coefficients. Note. Gender (1 = male, 2 = female). * *p* < 0.05. ** *p* < 0.01. *** *p* < 0.001.

**Table 1 ijerph-21-00355-t001:** Means, standard deviations, skewness and kurtosis for the main study variables.

	M	SD	Skewness	Kurtosis
Experienced contact	2.40	2.57	0.12	−1.19
Social dominance orientation	1.53	0.32	0.93	1.48
Beliefs in a just world	2.64	0.87	0.20	−0.34
Classical prejudice	1.58	0.43	0.95	0.72
Modern prejudice	2.10	0.43	−0.34	−0.48
Age	23.02	2.48	1.20	2.34

Note. Experienced contact ranges from –6 to +6.

**Table 2 ijerph-21-00355-t002:** Pearson’s bivariate correlations.

	1	2	3	4	5	6	7
1. Experienced contact	1.00						
2. Social dominance orientation	−0.35 ***	1.00					
3. Beliefs in a just world	−0.18 **	0.36 ***	1.00				
4. Classical prejudice	−0.28 ***	0.53 ***	0.37 ***	1.00			
5. Modern prejudice	0.00	0.12	0.40 ***	0.42 ***	1.00		
6. Age	−0.02	0.07	−0.06	0.07	0.02	1.00	
7. Gender	0.05	−0.07	0.03	−0.17 *	−0.01	−0.06	1.00

*Note*. Gender is a point bi-serial correlation (1 = male, 2 = female). Experienced contact ranges from –6 to +6. * *p* < 0.05. ** *p* < 0.01. *** *p* < 0.001.

## Data Availability

The data that support the findings of this study are available from the corresponding author upon reasonable request.

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
