# Peer review of "Classical and Modern Prejudice toward Individuals with Intellectual Disabilities: The Role of Experienced Contact, Beliefs in a Just World and Social Dominance Orientation"

_ijerph, 2024, doi:10.3390/ijerph21030355_

Round 1
Reviewer 1 Report
Comments and Suggestions for Authors
Thank you for the opportunity to review the article. The article is quite good. The theoretical introduction is sufficient. The research part is clearly described.
I have one essential comment about the research, which concerns the research sample:
- data on respondents' gender are missing (the authors are working with this factor)
- other data are missing (type of study, level of study, completion of a special subject that would deal with people with ID) - these factors can significantly influence the research results. The authors mention one sentence on this factor (study program) in "Limits," but it would be appropriate for the reader to improve the mention of this limitation.
If the authors did not ask them, it would be good to state this. A note is also related to how the sample of respondents was selected.
In this form, the results may appear unrepresentative and may lead to research bias.
Author Response
Dear Reviewer,
Thank you for sending us the reviewer’s comments on our paper, ‘Classical and Modern Prejudice toward Individuals with Intellectual Disabilities: The Role of Experienced Contact, Beliefs in a Just World, and Social Dominance Orientation’ which has helped us to strengthen our manuscript. We have now systematically revised the manuscript in line with each and every comment, and our responses to each of these can be found below. In the revised manuscript, our revisions have been highlighted in yellow. We hope the revised paper is now acceptable for publication in the International Journal of Environmental Research and Public Health.
Response to Reviewer 1
Reviewer’s comment: - data on respondents' gender are missing (the authors are working with this factor)
Authors’ response: In the revised manuscript, we have now included the gender of the participants.
Reviewer’s comment: other data are missing (type of study, level of study, completion of a special subject that would deal with people with ID) - these factors can significantly influence the research results. The authors mention one sentence on this factor (study program) in "Limits," but it would be appropriate for the reader to improve the mention of this limitation. If the authors did not ask them, it would be good to state this.
Authors’ response: Thank you for this comment. In the section “Participants and Procedures” of the current version of the manuscript, we provide information about the university degree program of the participants. Moreover, as requested, in the section “Limitations and Further Studies,” we have now improved this issue:
“Furthermore, additional studies should gather information about the students’ profiles about their university degree programs, focusing on educational subjects related to individuals affected by ID.”
Reviewer’s comment: A note is also related to how the sample of respondents was selected. In this form, the results may appear unrepresentative and may lead to research bias.
Authors’ response: As requested, we have now provided more information about how the sample was recruited:
“A sample of 224 Italian participants (62 males and 162 females) was recruited for this study using a snowball procedure. Initially, a class of university students was contacted and they were asked to fill in the online questionnaire and to share the link with their friends and mates.”
Reviewer 2 Report
Comments and Suggestions for Authors
The premise is interesting, and the path analysis is a particular strength of the manuscript, which explores serial mediators of the relationship between contact with people with ID and classical and modern prejudice. Most of my suggested edits focus on the Introduction...
1. While true, what is the point of noting that ID is included among other Developmental Disabilities? I was wondering if you were making a connection to the sentence that follows: ...despite advancements in understanding and accepting various disabilities...Were you intending to include other DD's in this sentence? Consider drawing a more explicit connection between these ideas.
2. Please say more about Sustainable Development Goals for the uninitiated reader (p. 1, line 41).
3. Give more explicit examples of classical and modern prejudice at the top of p. 2.
4. Bottom of p. 2, lines 88-99 should be moved to the end of the introduction, as these spell out your intended study goals. I think you can consolidate a lot of the material on pages 2 & 3 when you explain the BJW and SDO constructs and their supporting research vis-a-vis ID.
5. p. 4, line 164: "previous studies" needs a citation
6. Materials and Methods...224 participants should be written as the number. "a scientific field" (line 194) is traditionally written as Science, Technology, Engineering, and Mathematics (STEM). The universities in the next sentence should be redacted for peer review
7. p. 5, line 245 - "an optimal blend of the two is optimal"...Drop the first "optimal".
8. p. 6, line 268 - "We controlled for gender". Why? Please provide justification in your statement of hypotheses.
9. p. 6, line 279 - "supposed directions" should be written as "hypothesized directions"
10. p. 8, lines 350-359 strike me as very editorialized and not grounded in the results. Please consider rewriting with less speculative/opinionated language about the participants' beliefs.
Comments on the Quality of English LanguageThere are several places in the text where a citation number is standing in for a study or author's name (e.g., Fist sentence of introduction: According to [1], intellectual disabilities...). This first sentence should read, Schalock and colleagues...Please do a thorough proofread and make corrections wherever a citation number stands in for a study or author. Other places include but are not limited to: p. 2, line 79 [27]; p. 3, lines 131-143; p. 4 line 167 (Drawing upon [45] theory of ideology and prejudice); p. 5, lines 217, 241, 243, & 246.
Author Response
Dear Reviewer,
Thank you for sending us the reviewer’s comments on our paper, ‘Classical and Modern Prejudice toward Individuals with Intellectual Disabilities: The Role of Experienced Contact, Beliefs in a Just World, and Social Dominance Orientation’ which has helped us to strengthen our manuscript. We have now systematically revised the manuscript in line with each and every comment, and our responses to each of these can be found below. In the revised manuscript, our revisions have been highlighted in yellow. We hope the revised paper is now acceptable for publication in the International Journal of Environmental Research and Public Health.
Response to Reviewer 2
Reviewer’s comment: 1. While true, what is the point of noting that ID is included among other developmental disabilities? I was wondering if you were making a connection to the sentence that follows: ...despite advancements in understanding and accepting various disabilities...Were you intending to include other DD's in this sentence? Consider drawing a more explicit connection between these ideas.
Authors’ response: As requested, we have now tried to connect the two sentences:
“Despite the widespread deinstitutionalization of individuals with ID today, societal attitudes remain sluggish in shedding the negative labels of “hopeless cases” and “a problematic population” that have been attached to them for so long [3]. These social perceptions can significantly impact the level of inclusion and community involvement for individuals with ID, affecting their overall quality of life and well-being [4–6]. Therefore, it is crucial to grasp the prevailing social attitudes toward ID to pinpoint areas that require attention in awareness campaigns and educational initiatives [7]”.
Reviewer’s comment: 2. Please say more about Sustainable Development Goals for the uninitiated reader (p. 1, line 41).
Authors’ response: As requested, we have now improved the description of the SDG:
“The SDGs provide a framework for global cooperation and action to address the world’s most pressing challenges in a holistic and integrated manner. The SDGs are a set of 17 global objectives adopted by the United Nations in 2015 as part of the 2030 Agenda for Sustainable Development. Each SDG is interconnected, and progress in one goal often depends on progress in others. They build upon the successes and shortcomings of the Millennium Development Goals (MDGs) and aim to address a broader range of interconnected issues, including poverty, inequality, climate change, environmental degradation, peace, and justice.”
Reviewer’s comment: 3. Give more explicit examples of classical and modern prejudice at the top of p. 2.
Authors’ response: As requested, we have now provided more explicit examples of classical and modern prejudice:
“According to Sears [18] modern prejudice is characterized by three components: denial of continued discrimination, antagonism toward minority group demands, and resentment about special favours for minority groups. Therefore, while classical prejudices often centered around race, ethnicity, and religion (e.g., racial segregation, antisemitism), modern prejudices (e.g., paternalism) encompass a broader range of identities and characteristics, reflecting the complexities of contemporary societies [19].”
Reviewer’s comment: 4. Bottom of p. 2, lines 88-99 should be moved to the end of the introduction, as these spell out your intended study goals. I think you can consolidate a lot of the material on pages 2 & 3 when you explain the BJW and SDO constructs and their supporting research vis-a-vis ID.
Authors’ response: As requested, we have now moved the suggested lines of the text at the end of the “Introduction”.
Reviewer’s comment: 5. p. 4, line 164: "previous studies" needs a citation
Authors’ response: In the revised version of the manuscript, we underlined that the predictions were made based on the results of the discussed studies.
Reviewer’s comment: 6. Materials and Methods...224 participants should be written as the number. "a scientific field" (line 194) is traditionally written as Science, Technology, Engineering, and Mathematics (STEM). The universities in the next sentence should be redacted for peer review
Authors’ response: As requested, we have now applied all the suggested modifications.
Reviewer’s comment: 7. p. 5, line 245 - "an optimal blend of the two is optimal"...Drop the first "optimal".
Authors’ response: Thank you for the suggestion. We have now removed the first word.
Reviewer’s comment: 8. p. 6, line 268 - "We controlled for gender". Why? Please provide justification in your statement of hypotheses.
Authors’ response: As requested, we have now provided a justification in the “Data Analysis” section.
Reviewer’s comment: 9. p. 6, line 279 - "supposed directions" should be written as "hypothesized directions"
Authors’ response: Thank you for the suggestion. We have now corrected the sentence.
Reviewer’s comment: 10. p. 8, lines 350-359 strike me as very editorialized and not grounded in the results. Please consider rewriting with less speculative/opinionated language about the participants' beliefs.
Authors’ response: As requested, we have now rewritten the results:
“These findings are consistent with previous research linking SDO to prejudice against socially subordinate groups [51] such as people affected by ID [11,17]. This result suggests the complexity of prejudice and the need for nuanced approaches to understanding its underlying mechanisms.”
Reviewer’s comment: There are several places in the text where a citation number is standing in for a study or author's name (e.g., first sentence of introduction: According to [1], intellectual disabilities...). This first sentence should read, Schalock and colleagues... Please do a thorough proofread and make corrections wherever a citation number stands for a study or author. Other places include, but are not limited to: p. 2, line 79 [27]; p. 3, lines 131-143; p. 4 line 167 (Drawing upon [45] theory of ideology and prejudice); p. 5, lines 217, 241, 243, & 246
Authors’ response: As requested, we proofread the entire manuscript and revised all the citations.
Round 2
Reviewer 1 Report
Comments and Suggestions for Authors
Thank you very much for accepting the comments.